# The Relationship between Perfectionism and Social Anxiety: A Moderated Mediation Model

**DOI:** 10.3390/ijerph191912934

**Published:** 2022-10-10

**Authors:** Yuzheng Wang, Jing Chen, Xiaoshuo Zhang, Xiaoxiao Lin, Yabin Sun, Ning Wang, Jinyan Wang, Fei Luo

**Affiliations:** 1CAS Key Laboratory of Mental Health, Institute of Psychology, Chinese Academy of Sciences, 16 Lincui Road, Chaoyang District, Beijing 100101, China; 2Department of Psychology, University of Chinese Academy of Sciences, Beijing 100049, China; 3Shanghai Key Laboratory of Mental Health and Psychological Crisis Intervention, School of Psychology and Cognitive Science, East China Normal University, Shanghai 200062, China

**Keywords:** trait mindfulness, perfectionism, perceived stress, social anxiety, moderated mediation model

## Abstract

Social anxiety is one of the mental health problems associated with perfectionism. The present study investigated the possible mediation of perceived stress in the relationship between perfectionism and social anxiety, and whether this mediation depends on the level of trait mindfulness. A total of 425 college students (female: 82.9%; mean age: M = 19.90 ± 1.06 years old) completed the Multidimensional Perfectionism Scale (MPS), the Chinese Perceived Stress Scale (CPSS), the Interaction Anxiousness Scale (IAS), and the Five-Facet Mindfulness Questionnaire (FFMQ). After controlling for age and gender, the moderated mediation analysis suggested that perfectionism significantly and positively predicted social anxiety and that perceived stress mediated the link between perfectionism and social anxiety. In addition, the indirect effect of perfectionism on social anxiety was moderated by trait mindfulness. Specifically, the indirect effect was weaker among the individuals with a high level of mindfulness compared to those with a low level of mindfulness. The findings of this study suggest that trait mindfulness significantly moderates the indirect effect of perfectionism on social anxiety via perceived stress.

## 1. Introduction

Social anxiety is a common emotional experience caused by the extreme fear of negative evaluation of others in social situations [1]. When it reaches a certain degree of severity such that functioning is impaired, it is referred to as social anxiety disorder [2]. Individuals with high social anxiety tend to be overly nervous, shy, or uncomfortable in different social situations [3]. This will hinder the establishment and maintenance of individual relationships with others and society, damage the quality of life [4], and may lead to loneliness, depression, and even suicide [5,6]. Therefore, the purpose of this study was to further explore the influencing factors and mechanisms of social anxiety.

The cognitive model of social anxiety indicates that setting high standards for social performance is one of the important factors for individuals to produce anxiety in the process of social interaction [7]. In fact, setting high standards is one of the main manifestations of perfectionism. Perfectionism refers to individuals setting high standards and expectations for themselves and others, worrying about mistakes, and emphasizing that everything is impeccable [8]. Moreover, Frost et al. (1990) have reported that perfectionism is a trait of those who set high standards, strive to complete work perfectly, and critically evaluate themselves. According to the comprehensive psychological maintenance model of social anxiety disorder [9], overestimating social standards and underestimating their ability to meet social standards are two important factors that lead to individual social anxiety. Perfectionistic individuals tend to aim for high standards at different social stages (e.g., social expectations, in social situations, and afterwards) [10]. However, due to the high level of these standards, it is difficult for individuals to achieve them, and the pressure to reach their goals is likely to bring anxiety to individuals [10]. A large number of studies have also found that perfectionism is associated with many mental health problems, including social anxiety [4,11,12,13]. For example, Hewitt et al. (2003) have reported that perfectionists tend to gain the approval of others and be considered a perfect person. If the individual is not very good at interpersonal skills or has no confidence in their ability to perform perfectly, it is likely to lead to social anxiety. Previous study also found that perfect self-expression was associated with a high level of social anxiety [13]. Therefore, perfectionism is one of the important factors of social anxiety.

However, the mechanisms between perfectionism and social anxiety needs to be further investigated. Hewitt and Flett (2002) put forward the mechanism of stress generation when exploring the relationship between perfectionism and psychopathology [14]. This mechanism suggests that one of the reasons why perfectionist individuals are associated with psychopathology is that they perceive more stress due to their unique cognitive, emotional, or behavioral patterns in the process of interacting with the environment. Specifically, perfectionists set unrealistic standards when engaging in activities, making choices, or pursuing a goal, which virtually puts themselves in stressful events or situations [10]. Numerous studies also have suggested that individuals with high levels of perfectionism are more likely to perceive great stress in their lives [15,16,17]. In addition, the general vulnerability model of perfectionism indicates that individuals with a high level of perfectionism are prone to be faced with considerable emotional distress when experiencing stressful events, such as depression [18] and anxiety symptoms [19]. Therefore, perfectionism may be associated with perceived stress, which in turn predicts a high level of social anxiety.

Previous models and theoretical hypotheses have shown that perfectionism has a significant positive association with perceived stress [20,21]. However, perfectionism does not always lead to maladaptive outcomes [19]. Compared with the high standard itself, the perfectionist’s negative self-evaluation and their inability to accept failure are the root causes of emotional distress [22,23]. Therefore, it is likely that there are other variables (e.g., some trait of the individual) that affect the relationship between perfectionism and society anxiety. In the past few decades, trait mindfulness has received extensive attention in the fields of emotional and cognitive psychology [24,25]. Trait mindfulness refers to an ability to pay attention to the moment without judgment [26]. Moreover, mindfulness emphasizes keeping attention on the present experience with an open, curious, and accepting attitude [27], which also includes some unpleasant feelings or thoughts [28]. However, once the high standards of perfectionism are not met, individuals may have negative emotions or thoughts. Individuals tend to avoid coping with or try to change unpleasant thoughts and emotions [29], which ultimately may be enhanced (e.g., perception of more stress) [30]. Mindful individuals may cope with unpleasant emotions or thoughts with an open and acceptable attitude in order to obtain more meaningful experiences [31]. In addition, according to the stress buffer hypothesis of mindfulness [32], one of the reasons why mindfulness can improve individual mental health is that it can reduce reactivity to stressors. Thus, on the one hand, trait mindfulness can effectively relieve negative thoughts and emotions [33,34]; on the other hand, individuals with high levels of mindfulness are likely to experience less tension and threat when facing stressful situations. Therefore, trait mindfulness may attenuate the relationship between perfectionism and stress.

Based on the above literature [7,14,31], the purpose of this study was to explore the relationship between perfectionism and social anxiety as well as to establish a moderated mediation model (Figure 1). The hypotheses were as follows: (a) perfectionism would significantly and positively predict social anxiety; (b) the mediating role of perceived stress between perfectionism and social anxiety would be significant; (c) the indirect effect of perfectionism on social anxiety would be moderated by trait mindfulness.

## 2. Methods

### 2.1. Participants and Procedure

A total of 425 valid data were collected from a university in Weihai, Shandong Province, China. The participants were from the department of Higher Education Administration, and included 76 (17.9%) males and 349 (82.1%) females, who were aged between 17 and 26 years old [mean (M) = 19.90; standard deviation (SD) = 1.06]. This study was approved by the ethics committee of the author’s institution. The researcher explained to the participants the purpose and significance of the study as well as the matters requiring their attention in order to obtain their cooperation. All participants reviewed informed consent and filled out questionnaires anonymously. There were no special requirements and no exclusion for participation. The completion time of the questionnaire was about 15 min. The participants were asked to read the instructions carefully and were told that there were no right or wrong answers. They only needed to answer the questionnaires according to their own actual situation. Participants received ¥15 for compensation.

### 2.2. Measurement

#### 2.2.1. Multidimensional Perfectionism Scale (MPS)

Perfectionism was measured by multidimensional perfection scale (MPS) [35]. The Chinese version of this scale was developed by Dai (2010) [36]. This scale consists of two subscales: perfectionism high standard (15 items) and perfectionism adaptability (14 items). It consists of 29 items rated on a 5-point scale (1 = very much disagree, 2 = somewhat disagree, 3 = no opinion, 4 = somewhat agree, 5 = very much agree). The score of the high standard subscale indicates the tendency of an individual to strive for perfectionism. (e.g., “I am very concerned about my image in the eyes of others”). The score of the adaptability subscale indicates the degree of maladjustment caused by perfectionism (e.g., “I am often upset because I am afraid that I will not achieve my ideal goal”). The two subscales had a good internal consistency in the current study (high standard: α = 0.826; adaptability: α = 0.856).

#### 2.2.2. Chinese Perceived Stress Scale (CPSS)

The CPSS was introduced and revised by Yang (2003) from abroad [37]. It is used to measure the level of individual perceived stress and includes 14 items rated on a 5-point scale (1 = never, 2 = seldom, 3 = sometimes, 4 = often, 5 = always). The sum of each item score indicates the level of stress experienced by an individual (e.g., “In the last month, I am upset because of something that happened unexpectedly”). The internal consistency of this scale was good in the current study (α = 0.814).

#### 2.2.3. Interaction Anxiousness Scale (IAS)

The IAS was used to measure the tendency of an individual to experience social anxiety independent of behavior [38]. It is a single-dimensional survey of 15 items rated on a 5-point scale (1 = does not fit me at all, 5 = fits me perfectly; e.g., I usually feel uncomfortable with a group of people I do not know). The scale has good reliability and validity in college students [38]. In the current study, this scale demonstrated good internal consistency (α = 0.855). 

#### 2.2.4. Five-Facet Mindfulness Questionnaire (FFMQ)

The level of trait mindfulness was measured by the Chinese version of FFMQ, which was revised by Deng et al. (2011) [39] in the Chinese version. This questionnaire includes 39 items that were rated on a 5-point Likert-type scale ranging from 1 (never or very rarely true) to 5 (very often or always true). E.g., “I pay attention to sounds, such as clocks ticking, birds chirping, or cars passing.” A higher score indicates a higher level of individual mindfulness. In the current study, this questionnaire demonstrated a good internal consistency (α = 0.713).

### 2.3. Data Analysis

The analyses were conducted using SPSS 23.0 (IBM Corp, 2015) and Hayes’s PROCESS version 3.3 [40]. First, descriptive statistics were used to calculate the SD and mean levels of the main variables, and Pearson’s correlations were used to assess the relationships among perceived stress, perfectionism, trait mindfulness, and social anxiety. Then, all these variables were mean-centered prior to the analyses. Model 7 was used to test whether trait mindfulness could moderate the mediation models.

## 3. Results

### 3.1. Descriptive Statistics and Pearson’s Correlations

Table 1 shows the values of the mean and SD of the variables and Pearson’s correlations between the variables. The results suggested that perfectionism was positively associated with perceived stress and social anxiety. Meanwhile, trait mindfulness was negatively associated with perfectionism, perceived stress, and social anxiety. Furthermore, social anxiety was positively associated with perceived stress.

### 3.2. Perfectionism and Social Anxiety: A Moderated Mediation Model

Next, model 7 was used to analyze the moderated mediation model. As shown in Table 2, perfectionism significantly and positively predicted perceived stress (β = 0.255, *p* < 0.001) and social anxiety (β = 0.155, *p* < 0.001) after controlling for gender and age. In addition, perceived stress significantly and positively predicted social anxiety (β = 0.557, *p* < 0.001). Furthermore, the interaction between perfectionism and trait mindfulness significantly and negatively predicted perceived stress (β = −0.083, *p* = 0.017).

In order to visually show the moderating effect of trait mindfulness on perfectionism and perceived stress, the trait mindfulness score was divided into three levels: mean minus one standard deviation (M − 1SD), mean (M), and mean plus one standard deviation (M + 1SD), and the moderation figure was drawn (Figure 2). Simple slope analysis showed that among individuals with a high level of trait mindfulness (M + 1SD), the predictive effect of perfectionism on social anxiety was significant (β = 0.172, *p* < 0.001). Among individuals with a low level of trait mindfulness (M − 1SD), the predictive effect of perfectionism on social anxiety was increased (β = 0.337, *p* < 0.001). The standard total effect, direct effect, and indirect effect of perfectionism and social anxiety are shown in Table 3. And 0 was not included in the 95% confidence interval, indicating that the direct and indirect effects of perfectionism and social anxiety were significant.

## 4. Discussion

The current study explored the relationship between perfectionism and social anxiety by establishing a moderated mediation model. The results showed that trait mindfulness significantly moderated the indirect effect of perfectionism on social anxiety via perceived stress.

### 4.1. Perfectionism and Social Anxiety

Consistent with previous studies [10,11,12], perfectionism significantly and positively predicted social anxiety. The comprehensive psychological maintenance model of social anxiety disorder [9] suggests that overestimating social standards and underestimating their ability to meet social standards are two important factors that lead to individual social anxiety. However, high standards and negative self-evaluation are two typical characteristics of perfectionism [41]. There is a close relationship between perfectionism and social anxiety. Furthermore, Hewitt et al. (2003) have reported that perfectionists are likely to have the need to be perfect in interpersonal expression, that is, to gain the approval of others and be considered a perfect person. If the individual is not very good at interpersonal skills or has no confidence in their ability to perform perfectly, it is likely to lead to social anxiety. Previous study also found that perfect self-expression was associated with a high level of social anxiety [13]. Therefore, perfectionism significantly and positively predicted social anxiety.

### 4.2. The Mediating Role of Perceived Stress

This study found that perceived stress mediated the relationship between perfectionism and social anxiety. Specifically, a higher degree of perfectionism indicated that the individual would perceive more stress and would have an increased probability of an increased level of social anxiety. Thus, stress is the key factor in the relationship between perfectionism and social anxiety. The results of this study supported the mechanism of stress generation by perfectionism [14], which suggests that perfectionists feel more stress when interacting with the social environment because they set standards that are too high. Some studies have shown that the main sources of stress among perfectionists are strict estimation of their performance, great attention on their negative aspects, and few satisfying experiences [42]. Stress tends to damage the life satisfaction of perfectionists and is also a mediating variable between perfectionism and worry [43]. Consistent with the “general vulnerability model” of perfectionism proposed by Dunkley et al. (2014) [44], individuals with high levels of perfectionism and stress are particularly vulnerable to emotional distress [45]. Therefore, perceived stress is an important variable linking perfectionism and social anxiety in social processes.

### 4.3. The Moderation of Trait Mindfulness

This study found that trait mindfulness moderated the relationship between perfectionism and social anxiety. Specifically, among individuals with a low level of trait mindfulness, there was a significant and positive prediction of perfectionism on social anxiety. Meanwhile, among individuals with a high level of trait mindfulness, the prediction was weaker. With an increase in the level of trait mindfulness, the predictive effect of perfectionism on social anxiety was significantly weakened. Trait mindfulness buffered the effect of perfectionism on social anxiety. The mindful coping model of Garland et al. (2009) suggests that trait mindfulness may help individuals pay attention to their own dynamic process of awareness rather than the content of awareness when under a stressful situation, which will expand their scope of awareness and increase attention flexibility [46]. Thus, they are likely to make a positive reassessment and then have positive emotions and cognition. Individuals with a high level of trait mindfulness tend to reappraise irrational beliefs associated with perfectionism and negative self-evaluation to reduce stress. Previous studies also have demonstrated that mindfulness can effectively reduce rumination [33], some irrational beliefs [47,48], and negative emotional experiences [34]. Therefore, a high level of trait mindfulness helps to relieve the experience of stress among perfectionists. Future research can explore whether mindfulness practice related to stress reduction (e.g., MBSR) reduce the stress level of perfectionistic individuals and reduce social anxiety.

### 4.4. Limitations

This study has several limitations that must be addressed. First, one of the main limitations was that the data were cross-sectional; thus, causal conclusions could not be made. Second, variables were measured through self-reporting, which is likely to be influenced by other factors (e.g., social desirability biases) [49]. Therefore, future research should use multiple assessment methods for evaluation to make the results more convincing. Finally, the current study only considered the role of perceived stress and trait mindfulness between perfectionism and social anxiety. It is not known whether there are other mediating and moderating factors between them. Future studies should explore the effects of more factors on the relationship between perfectionism and social anxiety, such as the role of emotional regulation.

### 4.5. Implications and Future Research

Despite these limitations, the model constructed in this study explains to some extent the impact of perfectionism on social anxiety. Also, the current study makes some important contributions. The theory of stress generation mechanism between perfectionism and depression was studied in prior research [14,21]. The current study extends the theory to social anxiety and emphasizes the importance of trait mindfulness. Besides contributing to the existing theory, our findings have practical implications as well. For example, clinical perfectionism, which is defined as “the overdependence of self-evaluation on the determined pursuit of personally demanding, self-imposed, standards in at least one highly salient domain, despite adverse consequences” [50], is associated with psychopathology, such as social anxiety disorder (SAD). According to the model, mindfulness reduces the effect of perfectionism on stress, and that lower stress then leads to lower social anxiety. Future researchers can explore whether stress regulation and mindfulness training have therapeutic effects on perfectionist individuals’ interpersonal problems.

## 5. Conclusions

The current study found that perfectionistic individuals were more likely to perceive more stress and have an increased probability of an increased level of social anxiety. In addition, the results indicated that the predictive effect of perfectionism on social anxiety was significantly weakened with an increase in the level of trait mindfulness. Trait mindfulness was a protective factor between perfectionism and perceived stress. Intervention aimed at improving mindfulness and emotional regulation may help reduce level of stress and improve the social anxiety, particularly in individuals who experience high level perfectionism.

## Figures and Tables

**Figure 1 ijerph-19-12934-f001:**
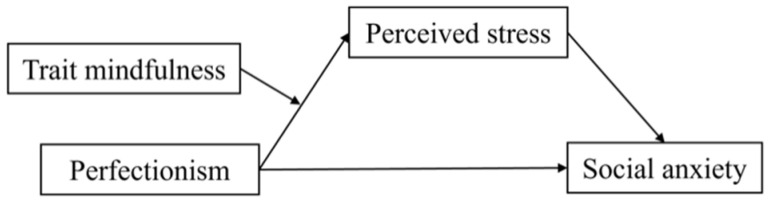
The hypothetical model.

**Figure 2 ijerph-19-12934-f002:**
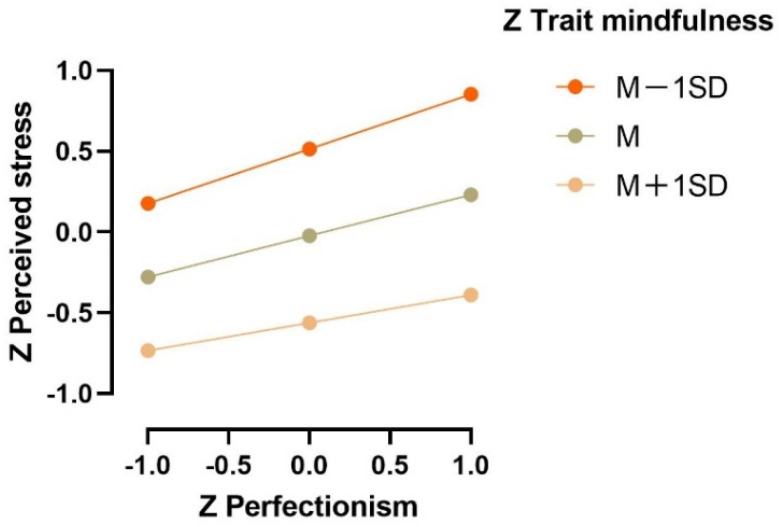
Simple slopes of the interaction effect: Trait mindfulness × Perfectionism (n = 425).

**Table 1 ijerph-19-12934-t001:** The mean (M), standard deviation (SD), and correlations of the variables (n = 425).

Variable	1	2	3	4	5	6
1 Perfectionism	-					
2 Perceived stress	0.39 ***	-				
3 Social anxiety	0.36 ***	0.62 ***	-			
4 Trait mindfulness	−0.29 ***	−0.59 ***	−0.49 ***	-		
5 Age	0.07	0.01	−0.03	0.03	-	
6 Gender	−0.10 *	0.02	0.11 *	0.75	-	-
M	45.26	38.54	44.72	123.31	19.90	-
SD	7.66	6.64	8.50	10.54	1.055	-

Note: *** *p* < 0.001; * *p* < 0.05.

**Table 2 ijerph-19-12934-t002:** The moderated mediating effect analysis of perfectionism on social anxiety (n = 425).

Variable	M: Perceived Stress			Y: Social Anxiety		
	β	SE	t	β	SE	t
Constant	−0.423	0.183	−2.306 *	−0.524	0.184	−2.852 **
Gender	0.219	0.098	2.230 *	0.288	0.099	2.913 **
Age	0.015	0.038	0.401	−0.035	0.038	−0.920
Perfectionism	0.255	0.040	6.454 ***	0.155	0.041	3.791 ***
Perceived stress				0.557	0.041	13.726 ***
Trait mindfulness	−0.538	0.039	−13.763 ***			
Perfectionism × Trait mindfulness	−0.083	0.035	−2.394 *			
R^2^	0.418			0.413		
F	60.212 ***			73.898 ***		

Note: * *p* < 0.05; ** *p* < 0.01; *** *p* < 0.001. SE, standard error.

**Table 3 ijerph-19-12934-t003:** Standard total effect, direct effect, and indirect effect of perfectionism and social anxiety (n = 425).

Outcome Variable	Trait Mindfulness	Effect	Bootstrap SE	Bootstrap 95% CI
Direct effect		0.155	0.041	[0.075, 0.235]
Indirect effect	M − 1SD	0.188	0.033	[0.128, 0.253]
	M	0.142	0.024	[0.099, 0.190]
	M + 1SD	0.096	0.030	[0.042, 0.159]

Note: SE, standard error; CI, confidence interval.

## Data Availability

The datasets generated and/or analyzed during the current study are available from the corresponding author upon reasonable request.

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
