# Peer review of "The Relationship between Perfectionism and Social Anxiety: A Moderated Mediation Model"

_ijerph, 2022, doi:10.3390/ijerph191912934_

Round 1

Reviewer 1 Report

Dear authors,

It is a good article and provides novel and topical results since it addresses mindfulness as a strategy. However, it is worth highlighting some aspects in which it would be convenient to improve. They are described below:

Introduction

- In the hypotheses it would be convenient to add a quote from a study that supports that hypothesis

Participants and Procedure

- It would be convenient to add if the administration of the questionnaire was voluntary and that the anonymity of the participants was respected.

- It would be convenient to add what the inclusion and exclusion criteria were.

Instruments

- It would be convenient to add how many items each dimension of the scales used is made up of and an example of an item.

Analysis of data

- The explanation of the data analysis is poor. Each analysis carried out must be better specified, as well as explaining what the criteria are to determine whether it is an adequate result or not based on the studies that speak of it.

Reviewer 2 Report

Thank you for allowing me to review this article titled "The relationship between perfectionism and social anxiety: A moderated mediation model". This paper begins with a brief review of the literature on the topic and proposes a simple moderated mediation model. The premise for this paper is strong and has the capacity to contribute to the field. I commend the authors on these strengths. However, there are a number of major issues that strongly limit the validity of the findings and conclusions. 

Major issues:

·         The choice of measures requires justification

o   Perfectionism measure - not consistent with the setup in the introduction – the authors discussed in the introduction that perfectionistic self-criticism was of note, but used a measure of perfectionistic standards.

§  Consider the two-factor model of perfectionism [see Stoeber, J., & Otto, K. (2006). Positive conceptions of perfectionism: Approaches, evidence, challenges. Personality and Social Psychology Review, 10(4), 295-319. https://doi.org/10.1207/s15327957pspr1004_2 ], these domains are discrete.

o   Interaction anxiety measure – the paper is setting up for social anxiety rather than non-clinical symptoms in a community sample (as you have), it would be important to see you comment on the clinical utility of this measure

·         The choice of analyses are not well justified. There is no justification for why the model was tested first as mediation only, and secondly with the moderating effects included. The second analysis (using model 7) means that the first (using model 4) is irrelevant.

o   I suggest removing all of the reporting of Model 4.

o   Repetition in reporting of results too – described in text, then in table, and then in figure form. I would suggest removing the table, and including this detail in the text and/or figure.

o   Results in table 1 – please include gender here

o   You report controlling for age and gender but have not reported these correlations to justify this choice

·         Discussion is too brief – condense the discussion of the results and consider further the implications of these findings for theory, research, and clinical practice.

o   Given your suggestions for lines 244 to 254, please make recommendations for how future research could test if this is true in the context of perfectionism and social anxiety.

o   Your final sentence (lines 273 -275) are too strong a conclusion based on your results – these should be framed a lot more tentatively, in considering implications for clinical practice.

o   Consider adding two separate headings for “future research” and  “conclusions” to provide structure and guide further additions to this discussion.

Minor issues:

·         Sentence one (line 30) requires a reference

·         Minor language/expression corrections – line 45 should read ‘perfectionistic individuals’ not perfectionist individuals.

·         Hypotheses – framing as social anxiety rather than symptoms associated with social anxiety. Given the lack of clinical sample or diagnostic measures, this needs to be rephrased

·         Language regarding mediation/indirect effects is conflicting – use language consistent with the analysis model (direct and indirect effects, rather than mediation).

·         Review formatting of figure 1 – moderating variable should be placed outside the triangle

·         Details about sample – was the sample within a faculty/school at the university (i.e., psychology students)? The gender balance indicates this is not a university-wide sample. Were students recruited through a participation pool? How were they compensated for their time?

·         Reporting p-values in text – ensure that you write the actual value (not <.05; e.g., page 5, line 183).

·         Content presented in the discussion should be reviewed first in the introduction – specifically, from line 209 to line 219.
